# SIRT1 Alleviates LPS-Induced IL-1β Production by Suppressing NLRP3 Inflammasome Activation and ROS Production in Trophoblasts

**DOI:** 10.3390/cells9030728

**Published:** 2020-03-16

**Authors:** Sumi Park, Jiha Shin, Jeongyun Bae, Daewon Han, Seok-Rae Park, Jongdae Shin, Sung Ki Lee, Hwan-Woo Park

**Affiliations:** 1Department of Cell Biology, Konyang University College of Medicine, Daejeon 35365, Korea; psm0503@konyang.ac.kr (S.P.); dixest0337@gmail.com (J.S.); 0518bjy@gmail.com (J.B.); eodnjs0319@naver.com (D.H.); shinjd@konyang.ac.kr (J.S.); 2Myunggok Medical Research Institute, Konyang University College of Medicine, Daejeon 35365, Korea; srpark@konyang.ac.kr; 3Department of Microbiology, Konyang University College of Medicine, Daejeon 35365, Korea; 4Department of Obstetrics and Gynecology, Konyang University Hospital, Daejeon 35365, Korea

**Keywords:** SIRT1, NLRP3 inflammasome, maternal inflammation, human trophoblasts, placenta, oxidative stress

## Abstract

Emerging evidence indicates that aberrant maternal inflammation is associated with several pregnancy-related disorders such as preeclampsia, preterm birth, and intrauterine growth restriction. Sirtuin1 (SIRT1), a class III histone deacetylase, is involved in the regulation of various physiopathological processes including cellular inflammation and metabolism. However, the effect of SIRT1 on the placental proinflammatory environment remains to be elucidated. In this study, we investigated the effect of SIRT1 on lipopolysaccharide (LPS)-induced NLRP3 inflammasome activation and its underlying mechanisms in human first-trimester trophoblasts (Sw.71 and HTR-8/SVneo cells). Treatment with LPS elevated SIRT1 expression and induced NLRP3 inflammasome activation in mouse placental tissues and human trophoblasts. Knockdown of SIRT1 enhanced LPS-induced NLRP3 inflammasome activation, inflammatory signaling, and subsequent interleukin (IL)-1β secretion. Furthermore, knockdown of NLRP3 considerably attenuated the increase of IL-1β secretion in SIRT1-knockdown cells treated with LPS. Moreover, SIRT1 inhibited LPS-induced NLRP3 inflammasome activation by reducing oxidative stress. This study revealed a novel mechanism via which SIRT1 exerts anti-inflammatory effects, suggesting that SIRT1 is a potential therapeutic target for the prevention of inflammation-associated pregnancy-related complications.

## 1. Introduction

Normal development of the placenta and proper functioning of trophoblasts are important for a healthy pregnancy. An adequate immune system response at the maternal–fetal interface is crucial for implantation and the preliminary stages of placentation [1]. Extravillous trophoblasts are highly proliferative cells that directly invade the uterine spiral arteries through the decidual stroma, and this process increases maternal blood flow to the placenta during early pregnancy [2,3]. Pathological inflammatory conditions such as intrauterine infection cause abnormal placental development, resulting in several pregnancy-related disorders, including preeclampsia, intrauterine growth restriction (IUGR), and preterm birth [4,5,6]. Trophoblasts exposed to inflammatory stimuli induce the secretion of proinflammatory cytokines such as tumor necrosis factor (TNF)-α and interleukin (IL)-6 [7,8], which contribute to defects in trophoblast function, placental dysfunction, and subsequent abnormal obstetric outcomes [9,10,11,12].

Self-oligomerizing scaffold proteins, also known as inflammasomes, assemble in response to pathogen-associated molecular patterns, including a TLR4 ligand; lipopolysaccharide (LPS); and damage-associated molecular patterns, including ATP. The NLRP3 inflammasome, one of the most important inflammasomes formed by the nucleotide-binding domain and leucine-rich repeat-containing (NLR) family members, is composed of an NLRP3 scaffold, an apoptosis-associated speck-like protein (ASC) adaptor, and caspase-1 [13]. In response to danger signals, the NLRP3 inflammasome initiates the activation of caspase-1 and processing of procytokines such as IL-1β [14,15]. Uric acid, palmitic acid, and nanosilica are known to activate the NLRP3 inflammasome signaling pathway in human trophoblast cells [16,17,18]. Mature IL-1β is a potent proinflammatory cytokine that is involved in inducing placental inflammatory response [19]. Therefore, aberrant IL-1β production in the placenta might be associated with abnormal NLRP3 inflammasome activation.

Sirtuin 1 (SIRT1), an NAD^+^-dependent protein deacetylase, plays a pivotal role in various cellular processes, including stem cell maintenance and differentiation and cellular metabolism [20,21,22]. SIRT1 deficiency results in developmental defects during mouse embryogenesis [23,24]. SIRT1 is localized in the syncytiotrophoblasts and cytotrophoblasts of the placenta [25]. Emerging evidence has demonstrated that SIRT1 not only negatively regulates the nuclear factor (NF)-κB signaling pathway but also mediates anti-inflammatory functions in several tissues [26,27,28,29]. However, whether SIRT1 regulates NLRP3 inflammasome activation in trophoblasts remains to be clarified.

In this study, we tested the hypothesis that SIRT1 is induced in response to LPS in trophoblasts and has a beneficial effect on placental inflammation. We also investigated the role of SIRT1 in LPS-induced NLRP3 inflammasome activation and its underlying mechanism in trophoblasts. We found that SIRT1 expression is induced in the placental tissues of LPS-treated mice as well as in LPS-treated trophoblasts. Knockdown of SIRT1 enhances LPS-induced NLRP3 inflammasome activation, inflammatory signaling, and IL-1β secretion via the increase in oxidative stress. Finally, we determined that LPS-induced SIRT1 suppresses IL-1β secretion in an NLRP3-dependent manner.

## 2. Materials and Methods

### 2.1. Reagents

Antibodies used for immunoblotting included SIRT1, ASC, and c-Myc from Santa Cruz Biotechnology (Dallas, TX, USA); SIRT1 from Abcam; NLRP3, Caspase-1, phospho-p65, and p65 from Cell Signaling Technology (Danvers, MA, USA); β-actin and α-tubulin from Developmental Studies Hybridoma Bank (Iowa City, IA, USA). LPS (*Escherichia coli* LPS, serotype 0127:B8), ATP, and *N*-acetylcysteine (NAC) were purchased from Sigma-Aldrich (St. Louis, MO, USA). Sources of the other reagents are indicated in the specified methods.

### 2.2. Animals and Treatments

Female C57BL/6 mice aged 9–10 weeks were purchased from Samtako (Osan, Korea). All animal studies were conducted in accordance with the Guidelines for the Care and Use of Laboratory Animals of the National Institutes of Health and approved by the Animal Ethics Committee of Konyang University. The mice were maintained in an environment with controlled temperature (20–25 °C) and humidity (50% ± 5), with a 12-h light/dark cycle, and were given free access to food and water for a period of 1 week before the study period. Female mice were mated with male mice, and the presence of vaginal plugs the next morning confirmed successful mating. Pregnant mice were intraperitoneally injected with LPS (100 µg/kg) or saline (vehicle) on gestation day 17.5. The pregnant mice were sacrificed at various time points (4, 6, 8, or 10 h) after LPS treatment, and the placentas were collected for molecular analysis.

### 2.3. Cell Culture and Treatments

Two human first trimester trophoblast cell lines, i.e., Sw.71 cells (a gift from Dr. Gil Mor, Yale University School of Medicine, New Haven, CT, USA) and HTR-8/SVneo cells (ATCC), were used in our experiments. The Sw.71 cells were cultured in Dulbecco’s modified Eagle’s medium (Welgene, Gyeongsan, Korea) containing 4500 mg/L d-glucose with l-glutamine, 10% fetal bovine serum (FBS, Welgene), and 100 U/mL penicillin–streptomycin (Welgene). The HTR-8/SVneo cells were cultured in RPMI 1640 (Welgene) medium containing 10% FBS, 100 U/mL penicillin-streptomycin, 4500 mg/L d-glucose, 10 mM HEPES, 2 mM l-glutamine, and 1 mM sodium pyruvate. All cultures were maintained in a humidified 5% CO_2_ atmosphere at 37 °C. Then, the Sw.71 and HTR-8/SVneo cells were incubated in the presence of 100 or 200 ng/mL LPS, or were treated with 100 or 200 ng/mL LPS for 12–48 h and then followed by 5 mM ATP treatment for 45 min. The same volume of phosphate-buffered saline (PBS) was used as the vehicle control.

### 2.4. Plasmids and Viral Production

The plasmid of pcDNA3-myc-SIRT1 was a gift from Dr. Hueng-Sik Choi (Chonnam National University, Gwangju, Korea). HEK293T cells were transfected with the following lentiviral constructs with packaging plasmids, using polyethylenimine reagent: sh-luciferase (sh-Luc), sh-SIRT1, and sh-NLRP3 constructs (Sigma-Aldrich, St. Louis, MO, USA). Lentiviral supernatants were collected and filtered at 48 and 72 h after transfection. The Sw.71 cells and HTR-8/SVneo cells were incubated for 2 days with lentiviral medium in the presence of 4 µg/mL polybrene.

### 2.5. Quantitative Real-Time Polymerase Chain Reaction (PCR)

Total RNA was extracted from placental tissues and from the Sw.71 cells using Trizol reagent (Takara, Shiga, Japan) according to the manufacturer’s instructions. Complementary DNA was synthesized using a Moloney murine leukemia virus reverse transcriptase (MMLV-RT, Promega, Madison, WI, USA) and random hexamers (BioFact, Daejeon, Korea). Quantitative real-time reverse-transcription PCR was performed in triplicate on samples with SYBR green real-time PCR master mix reagent (BioFact) using the QuantStudio 3 Real-time PCR System (Life Technologies, Inc., Carlsbad, CA, USA). The following primers were used: *mouse Sirt1* (NM_001159589); forward 5′-GATACCTTGGAGCAGGTTGC-3′, reverse 5′-CTCCACGAACAGCTTCACAA-3′; *mouse Nlrp3* (NM_001359638); forward 5′- AGCCTTCCAGGATCCTCTTC-3′, reverse 5′-CTTGGGCAGCAGTTTCTTTC-3′; *mouse Tnf-α* (NM_001278601); forward 5′-TCCCAGGTTCTCTTCAAGGGA-3′, reverse 5′-GGTGAGGAGCACGTAGTCGG-3′; *mouse Il-6* (NM_031168); forward 5′- TAGTCCTTCCTACCCCAATTTCC-3′, reverse 5′-TTGGTCCTTAGCCACTCCTTC-3′; *mouse Mcp-1* (NM_011333); forward 5′-CATCCACGTGTTGGCTCA-3′, reverse 5′-GATCATCTTGCTGGTGAATGAGT-3′; *mouse Il-1β* (NM_008361); forward 5′-TCTTTGAAGTTGACGGACCC-3′, reverse 5′-TGAGTGATACTGCCTGCCTG-3′; *mouse Gapdh* (NM_008084); forward 5′-AAGGTCATCCCAGAGCTGAA-3′, reverse 5′-CTGCTTCACCACCTTCTTGA-3′; *human NLRP3* (NM_001127462); forward 5′-GATCTTCGCTGCGATCAACAG-3′, reverse 5′-CGTGCATTATCTGAACCCCAC-3′; *human IL-8* (NM_000584); forward 5′-TTTTGCCAAGGAGTGCTAAAGA-3′, reverse 5′-AACCCTCTGCACCCAGTTTTC-3′; *human TNF-α* (NM_000594); forward 5′- GAGGCCAAGCCCTGGTATG-3′, reverse 5′-CGGGCCGATTGATCTCAGC-3′; *human IL-6* (NM_000600); forward 5′-ACTCACCTCTTCAGAACGAATTG-3′, reverse 5′-CCATCTTTGGAAGGTTCAGGTTG-3′; *human GAPDH* (NM_002046); forward 5′- TTGCCATCAATGACCCCTTCA-3′, reverse 5′-CGCCCCACTTGATTTTGGA-3′. The reaction was performed in a total volume of 20 µL per reaction. The reaction mixture included 2 μL of template cDNA, 0.5 μM of each primer, 10 μL 2 × SYBR green real-time PCR master mix, and sterile water. The amplification protocol started with 95 °C for 15 min, followed by 40 cycles at 95 °C for 15 s and 60 °C for 30 s. Relative mRNA expression was calculated from the comparative threshold cycle (Ct) values relative to mouse or human GAPDH.

### 2.6. Luciferase Reporter Assay

SIRT1 promoter constructs were generated by PCR amplification of the human SIRT1 promoter region (−1183 to −30 relative to the transcription start site) and were subcloned into the pGL3-basic vector (Promega, Madison, WI, USA) containing a firefly luciferase gene. The Sw.71 cells were transiently transfected with a pGL3-SIRT1 plasmid and a pRL-TK plasmid containing a Renilla luciferase gene as an internal control using polyethylenimine. At 48 h after transfection, the cells were treated with LPS or vehicle for 4 h. The activities of both luciferases were measured using the Dual-Luciferase Reporter System (Promega, Madison, WI, USA) according to the manufacturer’s instructions, and luminescent signals were detected using a GloMax 20/20 luminometer (Promega). For each well, the relative luciferase activity was normalized to the firefly luminescence/Renilla luminescence ratio.

### 2.7. Immunoblotting

The Sw.71 cells and HTR-8/SVneo cells were lysed in an ice-cold radioimmunoprecipitation assay buffer containing complete protease inhibitor cocktail (Roche, Basel, Switzerland). The lysates were incubated for 20 min on ice and centrifuged at 18,000 g for 15 min at 4 °C. The protein concentration was measured using the BCA Protein Assay (Pierce). The lysates were boiled in 1 × sodium dodecyl sulfate (SDS) LaemmLi sample buffer for 5 min, resolved using SDS-polyacrylamide gel electrophoresis, transferred onto polyvinylidene fluoride membranes (Millipore, Burlington, MA, USA), and probed with primary antibodies against SIRT1, NLRP3, p-p65, p65, ASC, Caspase-1, c-Myc, α-tubulin, or β-actin. After sample incubation with secondary antibodies conjugated with horseradish peroxidase, chemiluminescence signals were detected using the Fusion Solo System (Vilber Lourmat, Marne-la-Vallée, France). Densitometric analysis of the blots was performed using ImageJ software (National Institutes of Health, NIH, Bethesda, MD, USA), with which the background was removed for each band.

### 2.8. Immunocytochemistry

Sw.71 cells and HTR-8/SVneo cells grown on coverslips were rinsed with PBS and fixed with 4% paraformaldehyde (pH 7.4) for 15 min at room temperature. The cells were blocked in blocking solution for 1 h at room temperature and incubated with an anti-SIRT1 antibody (1:200) or an anti-ASC antibody (1:200) overnight at 4 °C in a humidified chamber. After washing, the cells were incubated with Alexa Fluor-conjugated secondary antibodies (Invitrogen, 1:500, Carlsbad, CA, USA) and mounted with ProLong Gold antifade reagent with 4′,6-diamidino-2-phenylindole (DAPI; Invitrogen, Carlsbad, CA, USA). Fluorescent images were obtained using a laser scanning confocal microscope (LSM 700, Carl Zeiss) or an epifluorescence-equipped microscope (DM2500, Leica, Wetzlar, Germany) and were processed using ImageJ software (NIH). The percentages of cells containing ASC specks relative to the total number of cells was calculated in five randomly chosen fields.

### 2.9. Immunohistochemistry

Mouse placental tissues were fixed in 10% neutral buffered formalin for 24 h, dehydrated, and embedded in paraffin. Paraffin-embedded sections were deparaffinized, rehydrated, and then treated for antigen retrieval. Endogenous peroxidase was quenched with 3% hydrogen peroxide. After blocking nonspecific antigen, the slides were then incubated with anti-SIRT1 antibody overnight at 4 °C, followed by incubation with biotinylated secondary antibody (Vector Laboratories, Burlingame, CA, USA). Antibodies were visualized with Streptavidin-HRP (BD Pharmingen, San Diego, CA, USA) using diaminobenzidine (Sigma-Aldrich, St. Louis, MO, USA). Hematoxylin was applied to visualize nuclei.

### 2.10. Detection of Reactive Oxygen Species (ROS)

The intracellular level of ROS was analyzed using the ROS-reactive fluorescent indicator 5-(and-6)-chloromethyl-2′,7′-dichlorodihydrofluorescein diacetate, acetyl ester (CM-H_2_DCFDA, Invitrogen, Carlsbad, CA, USA), according to the manufacturer’s instructions. The Sw.71 cells were seeded on a μ-Slide four-well chamber slide, treated with 5 µM of CM-H_2_DCFDA for 30 min, and washed with PBS. Samples were analyzed under a fluorescence microscope (Eclipse TS2, Nikon, Tokyo, Japan).

### 2.11. Enzyme-Linked Immunosorbent Assay (ELISA)

Cell supernatants collected from the Sw.71 cells were assayed for levels of the secreted inflammatory cytokine IL-1β using the DuoSet ELISA development kit (R&D Systems, Minneapolis, MN, USA) according to the manufacturer’s instructions. The optical density of the final colored reaction product was determined at 450 nm using an Epoch 2 microplate reader (Bio-Tek Instruments, Winooski, VT, USA).

### 2.12. Statistical Analysis

Results are presented as the mean ± standard error of mean (SEM). Unless mentioned otherwise, the data presented in the figure panels are representative of at least three independent experiments. The significance of differences between two experimental groups was determined using a two-tailed Student’s *t*-test. A *p* value less than 0.05 was considered statistically significant (* *p* < 0.05; ** *p* < 0.01; *** *p* < 0.001).

## 3. Results

### 3.1. SIRT1 Is Upregulated in the Placentas and Trophoblast Cells after LPS Exposure

To determine changes in SIRT1 expression in the mouse placenta by LPS-induced maternal inflammatory response, placentas were collected from mice exposed to LPS or saline. Placental SIRT1 mRNA expression was increased 8 h after LPS treatment (Figure 1A). Immunohistochemistry analysis confirmed that SIRT1 expression was increased in the nuclei of trophoblast in both the junctional and the labyrinth zones of the placentas of LPS-treated mice compared with that of the control placentas of saline-treated mice (Figure 1B). To investigate changes in the SIRT1 mRNA expression in trophoblasts during maternal inflammation, human first trimester extravillous trophoblasts (Sw.71 and HTR-8/SVneo) were treated with various LPS concentrations at various time points. Immunoblot analysis showed that LPS increased SIRT1 expression in the Sw.71 (Figure 1C) and HTR-8/SVneo cells (Appendix A) in a dose-dependent manner. The time courses of SIRT1 expression were further studied by treating the cells with LPS (200 ng/mL). The results showed that LPS increased the expression of SIRT1 in the Sw.71 (Figure 1D) and HTR-8/SVneo cells (Appendix A) in a time-dependent manner. Immunofluorescence staining was performed to determine the subcellular localization of SIRT1 in the Sw.71 and HTR-8/SVneo cells. The increased SIRT1 was localized exclusively to the nucleus in the Sw.71 cells after treatment with LPS, which differed from findings in control cells (Figure 1E). To understand the regulation of SIRT1 transcription in the LPS-treated Sw.71 cells, relative luciferase activity was assessed. The results showed that luciferase activity was significantly higher in the LPS-treated cells than in the untreated cells (Figure 1F).

### 3.2. Inflammation and NLRP3 Inflammasome Activation in the Placentas and Trophoblast Cells after LPS Exposure

To investigate the effects of LPS on the mRNA expression levels of NLRP3 inflammasome-related molecules, placentas from mice exposed to LPS were analyzed by quantitative real-time PCR. The results showed that placental NLRP3 mRNA expression was significantly increased 8 h after LPS treatment (Figure 2A). Proinflammatory gene expression in placentas from mice exposed to LPS was examined. The IL-6, TNF-α, MCP-1, and IL-1β mRNA levels were significantly elevated in the placentas of LPS-treated mice compared with control placentas from saline-treated mice (Figure 2B). Next, we performed an immunoblot analysis to investigate the effects of LPS on NLRP3 inflammasome expression in Sw.71 and HTR-8/SVneo cells. LPS treatment significantly increased NLRP3 inflammasome expression in Sw.71 (Figure 2C) and HTR-8/SVneo cells (Appendix A) in a dose-dependent manner. When the Sw.71 cells were treated with LPS (200 ng/mL) for various time periods, NLRP3 expression was substantially induced increased in a time-dependent manner (Figure 2D). Similar to its action on the Sw.71 cells, LPS treatment stimulated NLRP3 protein levels in HTR-8/SVneo cells (Appendix A). To investigate the NLRP3 inflammasome activation in Sw.71 and HTR-8/SVneo cells, the expressions of caspase-1 and ASC were evaluated upon stimulation with LPS and ATP by immunoblotting and immunofluorescence. This treatment increased caspase-1 activation but not the expression of the ASC (Figure 2E and Appendix A). Immunostaining for endogenous ASC showed that Sw.71 cells contained the ASC pyroptosome after stimulation with LPS and ATP (Figure 2F). We further determined whether NF-κB, a master regulator of inflammation, was activated in trophoblasts after LPS exposure. Immunoblot analysis demonstrated increased levels of phosphorylated NF-κB in Sw.71 cells (Figure 2G) and HTR-8/SVneo cells (Appendix A) at 1 h after LPS treatment.

### 3.3. SIRT1 Attenuates LPS-Induced Inflammation and NLRP3 Inflammasome Activation in Trophoblast Cells

To evaluate the regulatory effect of SIRT1 on LPS-induced NLRP3 inflammasome activation, lentiviral vectors expressing small interfering RNAs (shRNAs) for human SIRT1 were transduced to Sw.71 and HTR-8/SVneo cells. After the cells were transduced with sh-SIRT1-expression lentivirus, the expression of SIRT1 was remarkably reduced in the Sw.71 (Figure 3A) and HTR-8/SVneo cells (Appendix A). Knockdown of SIRT1 obviously enhanced the NLRP3 mRNA and protein levels after LPS treatment (Figure 3A,B and Appendix A). To further confirm these results, the IL-1β levels in the culture supernatants were analyzed using ELISA. The results showed that secreted IL-1β levels were significantly increased in LPS- and ATP-treated SIRT1 knockdown cells compared with those in LPS- and ATP-treated control knockdown cells (Figure 3C). Correspondingly, knockdown of SIRT1 also increased pro-IL-1β expression in LPS-treated Sw.71 cells (Figure 3D). Consistent with its effect on IL-1β production, knockdown of SIRT1 increased caspase-1 activation (Figure 3E) and ASC pyroptosome formation (Figure 3F). These data demonstrate that deficiency of SIRT1 augments NLRP3 inflammasome activation and subsequently the maturation of IL-1β. Next, we investigated the effect of overexpression of SIRT1 on LPS-induced NLRP3 inflammasome activation in Sw.71 cells. The results showed that overexpression of SIRT1 reduced the expression of NLRP3 protein and caspase-1 activation (Appendix A). These results suggest that SIRT1 attenuates NLRP3 inflammasome activation.

NF-κB mediates the priming signal of NLRP3 inflammasome activation and upregulates the transcriptional expression of NLRP3, as well as of pro-IL-1β [30,31]. To investigate the effects of knockdown of SIRT1 on the NF-κB pathway in Sw.71 and HTR-8/SVneo cells, we measured the phosphorylation levels of NF-κB. Immunoblot analysis of p65 NF-κB showed that the LPS-induced phosphorylation level of p65 NF-κB was higher in the SIRT1 knockdown groups than in the control knockdown groups in the Sw.71 (Figure 4A) and the HTR-8/SVneo cells (Figure 4B). Moreover, we examined the expression of proinflammatory cytokine genes, including IL-6, IL-8, and TNF-α in SIRT1 knockdown Sw.71 cells using quantitative real-time PCR. The results showed that the IL-6, IL-8, and TNF-α levels were significantly augmented by the knockdown of SIRT1 in Sw.71 cells (Figure 4C).

### 3.4. SIRT1 Suppresses LPS-Induced IL-1β Secretion in Trophoblast Cells in a NLRP3-Dependent Manner

To examine whether SIRT1 inhibition of IL-1β secretion was dependent on NLRP3 inflammasomes in trophoblasts treated with LPS and ATP, lentiviral vectors expressing shRNAs for human NLRP3 were transduced to Sw.71 cells. After the cells were transduced with sh-NLRP3-expression lentivirus, the NLRP3 expression was remarkably decreased in the Sw.71 cells (Figure 5A). Next, the IL-1β level in the culture supernatants was determined using ELISA. The lentivirus-mediated knockdown of NLRP3 significantly abrogated the increase of IL-1β secretion in the SIRT1 knockdown cells treated with LPS and ATP (Figure 5B). It is known that NLRP3 inflammasome regulates the NF-κB pathway as well as induces multiple cytokine genes [32,33]. We further investigated the effect of NLRP3 inflammasome on the activation of NF-κB in SIRT1 knockdown trophoblasts. Notably, LPS treatment in the SIRT1 knockdown cells elevated the phosphorylation of p65 NF-κB, which was abolished by lentivirus-mediated knockdown of NLRP3 (Figure 5C). We further examined the effects of the NLRP3 inflammasome on the expression of proinflammatory cytokine genes. LPS treatment in the SIRT1 knockdown cells induced IL-6, IL-8, and TNF-α mRNA expression, which was markedly abrogated by lentivirus-mediated knockdown of NLRP3 (Figure 5D). These findings suggest that the regulatory effect of SIRT1 on inflammatory response is associated with suppression of the NLRP3 inflammasome activation in trophoblasts.

### 3.5. SIRT1 Inhibits LPS-Induced NLRP3 Inflammasome in Trophoblast Cells Via Reducing Oxidative Stress

Various types of cellular stress stimuli trigger the generation of ROS, which may induce activation of the NLRP3 inflammasome through oxidative stress [34,35,36]. We examined whether LPS treatment induced oxidative stress in the Sw.71 cells. The CM-H_2_DCFDA fluorescence levels showed that LPS treatment stimulated ROS production (Figure 6A). Next, we tested whether the oxidative stress induced by LPS treatment was augmented by lentivirus-mediated knockdown of SIRT1. The results showed that ROS production induced by LPS treatment was significantly elevated by lentivirus-mediated knockdown of SIRT1 (Figure 6B). We further investigated the effect of overexpression of SIRT1 on LPS-induced oxidative stress. Relative to control cells, Sw.71 cells overexpressing SIRT1 had a lower level of CM-H_2_DCFDA fluorescence (Appendix A). Next, we sought to determine whether the antioxidant NAC could diminish the augmented NLRP3 inflammasome activation in LPS-treated SIRT1 knockdown cells. Immunoblot analysis showed that NAC treatment stopped the LPS-induced increase in NLRP3 levels in the SIRT1 knockdown cells (Figure 6C). Furthermore, the release of IL-1β was measured using ELISA. NAC treatment also decreased the IL-1β secretion in SIRT1 knockdown cells treated with LPS and ATP (Figure 6D). Subsequently, the effect of NAC on the activation of NF-κB in the SIRT1 knockdown trophoblasts was investigated. As shown in Figure 6E, NAC treatment decreased the phosphorylation of p65 NF-κB. These data suggest that SIRT1 attenuated the NLRP3 inflammasome activation that was induced by oxidative stress.

## 4. Discussion

Placental inflammation due to infectious or noninfectious causes has been linked with various adverse pregnancy outcomes, including preterm delivery, stillbirth, and IUGR. These inflammatory processes result in placental dysfunction and the release of several inflammatory mediators [37]. LPS, the major component of the outer membrane of Gram-negative bacteria, increases the production of various inflammatory cytokines and chemokines and induces innate immunity at the maternal–fetal interface [38,39]. Maternal and fetal adverse outcomes are associated with the release of various proinflammatory mediators such as cytokines and chemokines from the placenta. Elevated IL-1β production might be related to infection- or inflammation-induced spontaneous term parturition and preterm birth [40]. Recent studies have demonstrated a relationship between the maturation of IL-1β and the NLRP3 inflammasome pathway in human trophoblasts [41,42], but the molecular mechanisms underlying LPS-induced NLRP3 inflammasome activation are not completely understood. In the present study, we demonstrated that LPS treatment induces placental SIRT1 expression. The present study demonstrated that LPS treatment induces placental SIRT1 expression. Furthermore, LPS- and ATP-mediated NLRP3 inflammasome activation results in increases in IL-1β production in trophoblasts. Importantly, SIRT1 suppresses NLRP3 inflammasome activation and inflammation via reducing the oxidative stress in trophoblasts (Figure 7).

Although SIRT1 is expressed in all trophoblast layers of the placenta and is important for proper trophoblast differentiation and placental development [43], the relationship between LPS-associated placental inflammation and SIRT1 expression has not been studied previously. Recent studies showed that single nucleotide polymorphisms (SNPs) in the SIRT1 gene are associated with carotid atherosclerosis, major depressive disorder, age-related macular degeneration, and severe obesity [44,45,46,47]. However, it is not known whether SNPs in the SIRT1 gene have an influence on the NLRP3 inflammasome-mediated inflammatory diseases. Therefore, further studies are needed to investigate the possibility of genetic association between variants in the SIRT1 loci and inflammatory conditions. Previous studies demonstrated that SIRT1 regulates the inflammatory response in other cell types, such as adipocytes and endothelial cells [29,48,49]. Our study showed that LPS-induced SIRT1 expression is localized predominantly in the nucleus of trophoblasts. We also provided evidence on inhibition of LPS-mediated inflammation with SIRT1 by using a knockdown approach. The immunoblot analysis for p65 NF-κB identified an inhibitory effect of SIRT1 on the activation of NF-κB. We also found that SIRT1 knockdown resulted in a marked increase in the NLRP3 protein level in the Sw.71 and HTR-8/SVneo cells stimulated with LPS. These results suggest that SIRT1 functions in the placenta may not only be involved in the inflammatory response but also affect the NLRP3 inflammasome activation. Consistent with this notion, anti-inflammatory effects of SIRT1 are associated with regulation of the NLRP3 inflammasome in vascular endothelial cells and the epidermis [48,50]. Increasing evidence demonstrates that SIRT1 activators such as resveratrol and SRT1720 can inhibit NF-κB activation as well as inflammatory pathways [51,52,53]. Furthermore, SIRT1 activators inhibited the increase in NLRP3 and IL-1β production in mesenchymal stem cells and vascular endothelial cells [53,54]. Our present study showed that overexpression of SIRT1 inhibited the expression of NLRP3 protein and caspase-1 activation. Therefore, these results indicate that SIRT1 activators might represent suitable therapeutic options for the treatment of inflammation-associated pregnancy complications.

Although SIRT1 has been shown to regulate the NLRP3 inflammasome, its molecular mechanism remains to be identified. NLRP3 inflammasome activation could be induced by ROS [55]. Hypoxia can cause placental tissue damage, including preeclampsia and IUGR, via the production of oxidative stress [56]. In the present study, intracellular ROS amounts were found to be elevated in Sw.71 cells after LPS treatment. In addition, ROS production induced by LPS treatment was aggravated by knockdown of SIRT1, indicating that SIRT1 ameliorates oxidative stress. We have observed that NAC treatment inhibited an increase in NLRP3 levels, as well as the phosphorylation of p65 NF-κB in SIRT1 knockdown cells treated with LPS. These results suggest that SIRT1 inhibits the NLRP3 inflammasome activation through inhibiting oxidative stress. Previous studies have shown bidirectional crosstalk between SIRT1 and nuclear factor erythroid 2-related factor 2 (Nrf2) in human renal proximal tubular and glomerular mesangial cells [57,58]. The upregulation of SIRT1 activates Nrf2 through the inhibition of p53 or activation of AMPK [59,60,61]. In addition, it is known that in ischemic injury Nrf2 suppresses NLRP3 inflammasome activation through regulating the thioredoxin-interacting protein (TXNIP) complex [62]. Thus, Nrf2 may be linked with negative regulation of the NLRP3 inflammasome by SIRT1; however, further experimental studies are needed to explore this concept.

## 5. Conclusions

In summary, the results of this study provide evidence for a potential role of SIRT1 in LPS-induced NLRP3 inflammasome activation in trophoblasts by showing that the knockdown of SIRT increased LPS-induced inflammatory signaling, IL-1β secretion, and NLRP3 expression. SIRT1 also ameliorates oxidative stress caused by LPS treatment. Our study elucidates a novel mechanism by which SIRT1 exerts anti-inflammatory effects and defines SIRT1 as a therapeutic target in the prevention of inflammation-associated pregnancy complications.

## Figures and Tables

**Figure 1 cells-09-00728-f001:**
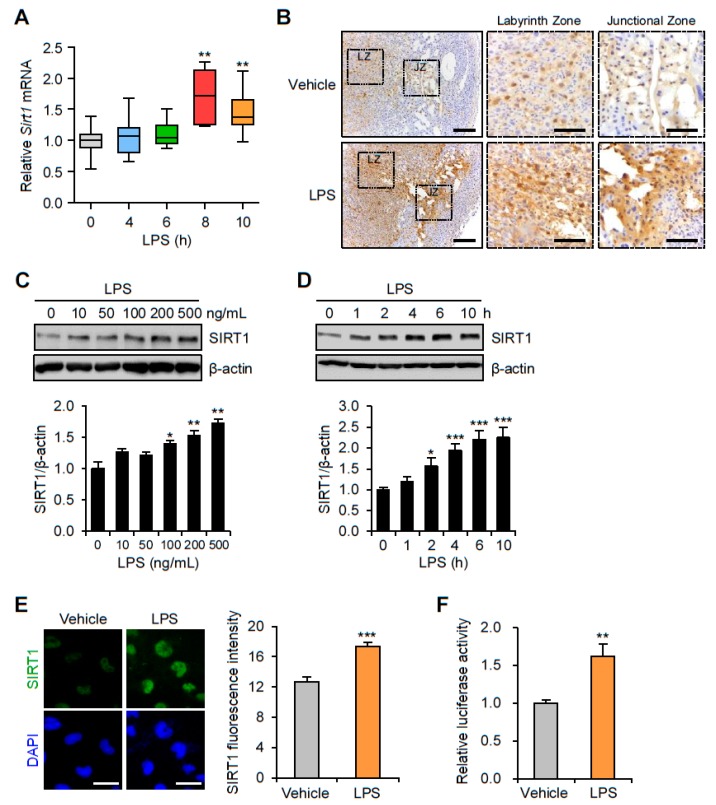
Expression of SIRT1 in placenta of mice and in human trophoblasts exposed to lipopolysaccharide (LPS). (**A**) C57BL/6 pregnant mice were injected intraperitoneally with LPS (100 μg/kg) on gestation day 17.5. After the indicated number of hours, placentas were harvested and analyzed by real-time PCR for expression of SIRT1 (*n* = 5–11). (**B**) Immunohistochemistry analysis of SIRT1 in placental tissues from mice exposed to LPS or vehicle for 8 h. Nuclei were stained with hematoxylin. Boxed areas are magnified in middle panels. LZ and JZ represent the labyrinth zone and the junctional zone respectively. Scale bar, 200 μm; inset, 100 μm. (**C**) Sw.71 cells were treated with the indicated concentration of LPS for 4 h. Cell lysates were immunoblotted with an anti-SIRT1 antibody. β-Actin served as a loading control. Band intensities were quantified and normalized over the β-actin values. (**D**) Sw.71 cells were treated with 200 ng/mL LPS for the indicated time periods. Cell lysates were immunoblotted with an anti-SIRT1 antibody. β-Actin served as a loading control. Band intensities were quantified and normalized over the β-actin values. (**E**) Immunofluorescence staining of SIRT1 (green) in Sw.71 cells treated with LPS (200 ng/mL) or vehicle for 4 h. Nuclei were stained with DAPI (blue). Scale bars, 40 μm. (**F**) Sw.71 cells that were transfected with firefly luciferase constructs containing an SIRT1 promoter sequence. At 48 h after transfection, cells were treated with LPS (100 ng/mL) for 4 h. Firefly luciferase activity was measured and normalized to the Renilla luciferase activity. Data are shown as mean ± SEM. Results are representative of at least three independent experiments. * *p* < 0.05; ** *p* < 0.01; *** *p* < 0.001 (Student’s *t*-test).

**Figure 2 cells-09-00728-f002:**
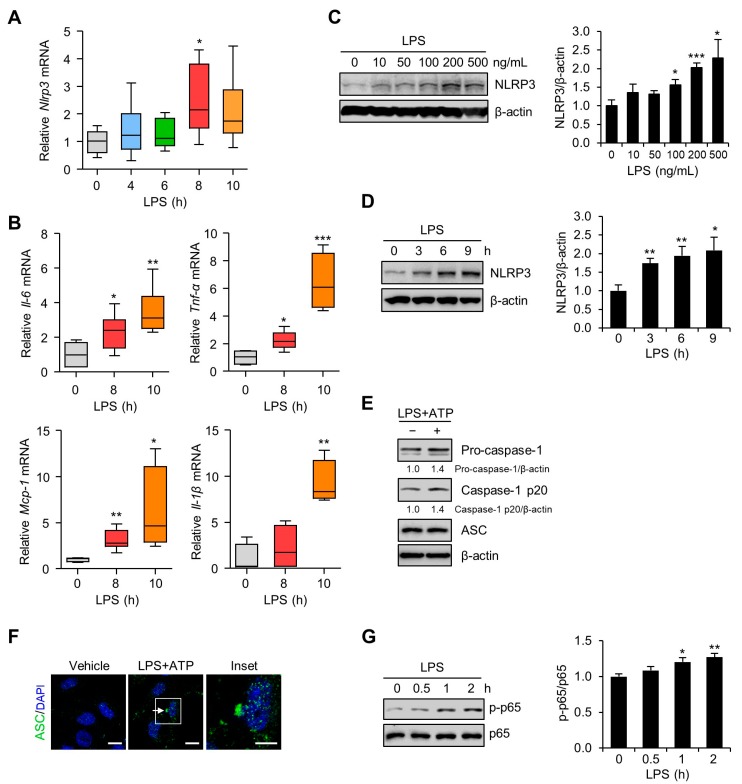
NLRP3 inflammasome activation in placentas of mice and in human trophoblasts exposed to LPS. (**A**,**B**) C57BL/6 pregnant mice were injected intraperitoneally with LPS (100 μg/kg) on gestation day 17.5. After the indicated number of hours, placentas were harvested and analyzed using real-time PCR for the expression of NLRP3, IL-6, TNF-α, MCP-1, and IL-1β (*n* = 4–7). (**C**) Sw.71 cells were treated with the indicated concentration of LPS for 4 h. Cell lysates were immunoblotted with an anti-NLRP3 antibody. (**D**,**G**) Sw.71 cells were treated with 200 ng/mL LPS for the indicated time periods. Cell lysates were immunoblotted with anti-NLRP3 (**D**) and anti-p-p65 NF-κB, and anti-p65 NF-κB (**G**) antibodies. (**E**) Sw.71 cells were treated with LPS (100 ng/mL) for 24 h and then followed by ATP (5 mM) treatment for 45 min, or were treated with vehicle. Cell lysates were immunoblotted with anti-caspase-1 and anti-ASC (apoptosis-associated speck-like protein) antibodies. β-Actin or p65 served as a loading control. Band intensities were quantified and normalized over the β-actin or p65 values. (**F**) Immunofluorescence staining of ASC (green) in Sw.71 cells treated with LPS (100 ng/mL) for 24 h and then followed by ATP (5 mM) treatment for 45 min, or treated with vehicle. The white arrow represents ASC specks. Nuclei were stained with DAPI (blue). Scale bars, 20 μm; inset, 10 μm. Data are shown as mean ± SEM. Results are representative of at least three independent experiments. * *p* < 0.05; ** *p* < 0.01; *** *p* < 0.001 (Student’s *t*-test).

**Figure 3 cells-09-00728-f003:**
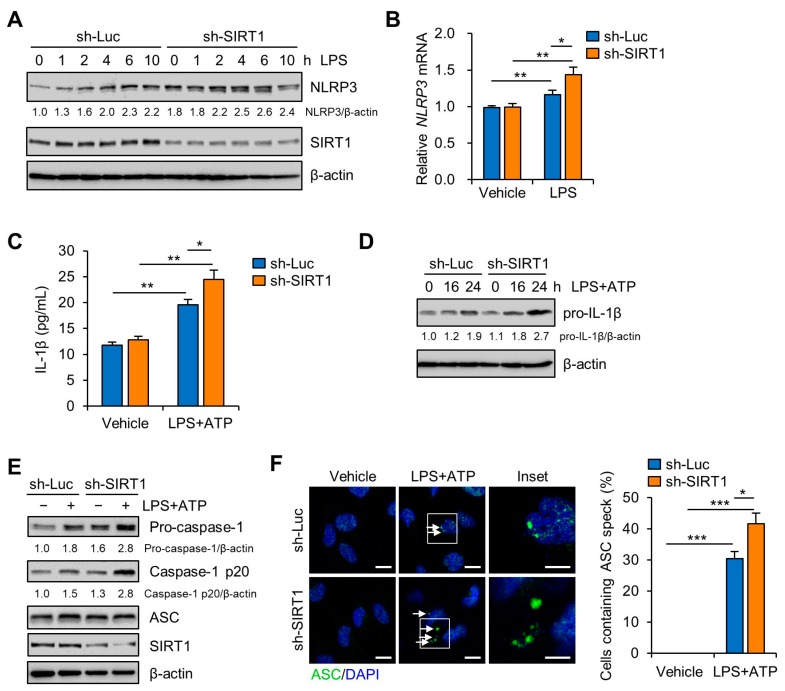
SIRT1 knockdown enhances LPS-induced NLRP3 inflammasome activation in trophoblasts. (**A**) Sw.71 cells were infected with lentiviruses expressing shRNAs targeting luciferase (sh-Luc) or SIRT1 (sh-SIRT1) and then treated with LPS (200 ng/mL) for the indicated time. Cell lysates were immunoblotted with anti-NLRP3 and anti-SIRT1 antibodies. (**B**) Relative mRNA expression of NLRP3 in Sw.71 cells infected with sh-Luc or sh-SIRT1 lentivirus and treated LPS (200 ng/mL) for 12 h. (**C**) Conditioned media were collected from Sw.71 cells infected with sh-Luc or sh-SIRT1 lentivirus and then treated or not (vehicle) with LPS (200 ng/mL) for 24 h, followed by ATP (5 mM) treatment for 45 min. Levels of IL-1β in the conditioned media were quantified by ELISA. (**D**) Sw.71 cells were infected with sh-Luc or sh-SIRT1 lentivirus and then treated or not (vehicle) with LPS (200 ng/mL) for the indicated time, followed by ATP (5 mM) treatment for 45 min. Cell lysates were immunoblotted with anti-IL-1β antibody. (**E**,**F**) Sw.71 cells were treated with LPS (100 ng/mL) for 24 h and then followed by ATP (5 mM) treatment for 45 min, or were treated with vehicle. (**E**) Cell lysates were immunoblotted with anti-caspase-1, anti-ASC, and anti-SIRT1 antibodies. (**F**) Immunofluorescence staining of ASC (green). The white arrow represents ASC specks. Nuclei were stained with DAPI (blue). Scale bars, 20 μm; inset, 10 μm. Endogenous ASC specks were quantified. β-Actin served as a loading control. Numbers below the immunoblot bands indicate the fold changes normalized to the control bands. Data are shown as mean ± S.E.M. Results are representative of at least three independent experiments. * *p* < 0.05; ** *p* < 0.01; *** *p* < 0.001 (Student’s *t*-test).

**Figure 4 cells-09-00728-f004:**
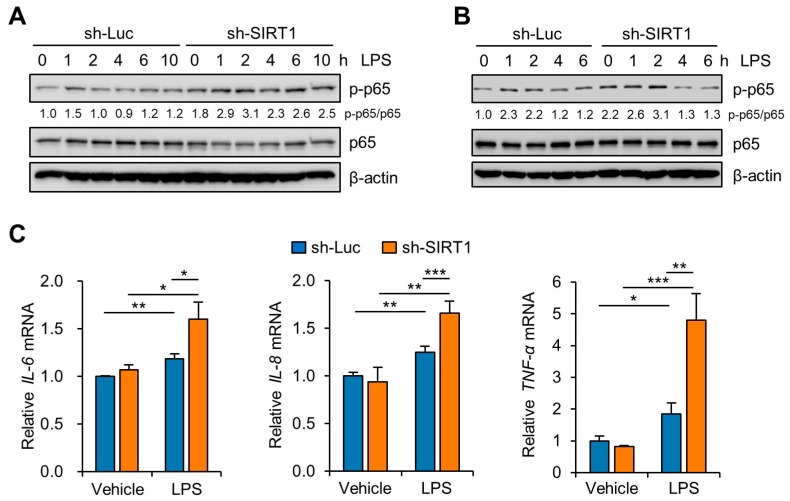
SIRT1 knockdown enhances LPS-induced inflammatory response in trophoblasts. (**A**,**B**) Sw.71 cells (**A**) and HTR-8/SVneo cells (**B**) were infected with lentiviruses expressing shRNAs targeting luciferase (sh-Luc) or SIRT1 (sh-SIRT1) and were then treated with 200 ng/mL LPS for the indicated time. Cell lysates were immunoblotted with anti-p-p65 and anti-p65 antibodies. β-Actin served as a loading control. Numbers below the immunoblot bands indicate the fold changes normalized to the p65 bands. (**C**) Relative mRNA expression of IL-6, IL-8, and TNF-α in Sw.71 cells infected with sh-Luc or sh-SIRT1 lentivirus and treated with 200 ng/mL LPS for 6 h. Data are shown as mean ± SEM. Results are representative of at least three independent experiments. * *p* < 0.05; ** *p* < 0.01; *** *p* < 0.001 (Student’s *t*-test).

**Figure 5 cells-09-00728-f005:**
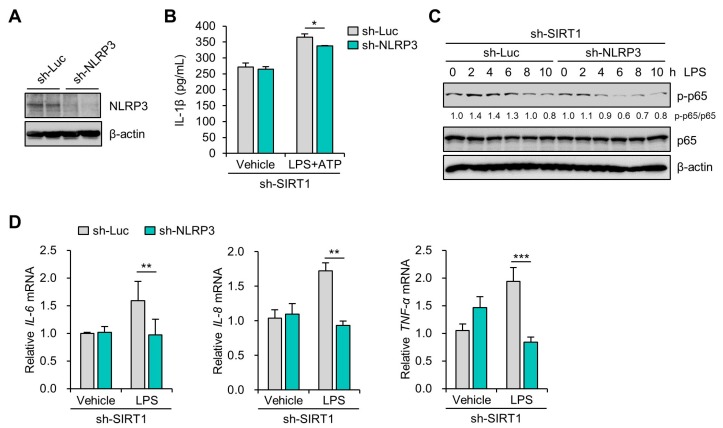
SIRT1 knockdown increases LPS-induced IL-1β secretion in an NLRP3-dependent manner. (**A**) Sw.71 cells were infected with lentiviruses expressing shRNAs targeting luciferase (sh-Luc) or NLRP3 (sh-NLRP3). Cell lysates were immunoblotted with anti-NLRP3 antibody. β-Actin served as a loading control. (**B**) Conditioned media were collected from Sw.71 cells infected with double sh-SIRT1/sh-Luc or sh-SIRT1/sh-NLRP3 lentivirus and then treated with 200 ng/mL LPS or vehicle for 48 h, with the addition of 5 mM ATP for the last 45 min. Levels of IL-1β in the conditioned media were quantified by ELISA. (**C**) Sw.71 cells were infected with double sh-SIRT1/sh-Luc or sh-SIRT1/sh-NLRP3 lentivirus and then treated with 200 ng/mL LPS for the indicated time. Cell lysates were immunoblotted with anti-p-p65 and anti-p65 antibodies. β-Actin served as a loading control. Numbers below the immunoblot bands indicate the fold changes normalized to the p65 bands. (**D**) Relative mRNA expression of IL-6, IL-8, and TNF-α in Sw.71 cells infected with double sh-SIRT1/sh-Luc or sh-SIRT1/sh-NLRP3 lentivirus and then treated with 200 ng/mL LPS for 6 h. Data are shown as mean ± SEM. Results are representative of at least three independent experiments. * *p* < 0.5; ** *p* < 0.01; *** *p* < 0.001 (Student’s *t*-test).

**Figure 6 cells-09-00728-f006:**
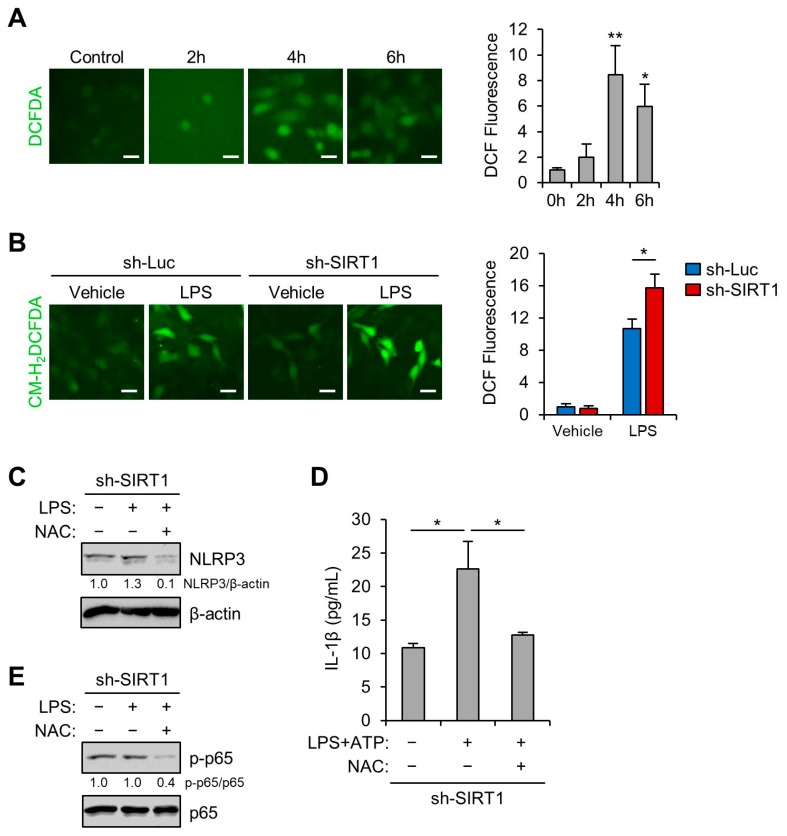
SIRT1 knockdown enhances LPS-induced NLRP3 inflammasome activation by reducing oxidative stress. (**A**) Sw.71 cells were treated with 200 ng/mL LPS for the indicated time. Reactive oxygen species (ROS) levels were analyzed using CM-H_2_DCFDA. Scale bars, 20 μm. (**B**) Sw.71 cells were infected with lentiviruses expressing shRNAs targeting luciferase (sh-Luc) or SIRT1 (sh-SIRT1) and then treated with 200 ng/mL LPS for 4 h. ROS levels were analyzed using CM-H_2_DCFDA. Scale bars, 40 μm. (**C**,**E**) Sw.71 cells were infected with sh-SIRT1 lentivirus and then treated with 200 ng/mL LPS or vehicle, each in the presence or absence of 5 mM *N*-acetylcysteine (NAC). Cell lysates were immunoblotted with anti-NLRP3, anti-p-p65, and anti-p65 antibodies. β-Actin or p65 served as the loading control. Numbers below the immunoblot bands indicate fold changes normalized to the control bands. (**D**) Conditioned media were collected from Sw.71 cells infected with sh-SIRT1 lentivirus and then treated with 200 ng/mL LPS alone or with 5 mM NAC for 34 h, with the addition of 5 mM ATP for the last 45 min. Levels of IL-1β in the conditioned media were quantified using ELISA. Data are indicated as mean ± SEM. Results are representative of at least three independent experiments. * *p* < 0.05; ** *p* < 0.01 (Student’s *t*-test).

**Figure 7 cells-09-00728-f007:**
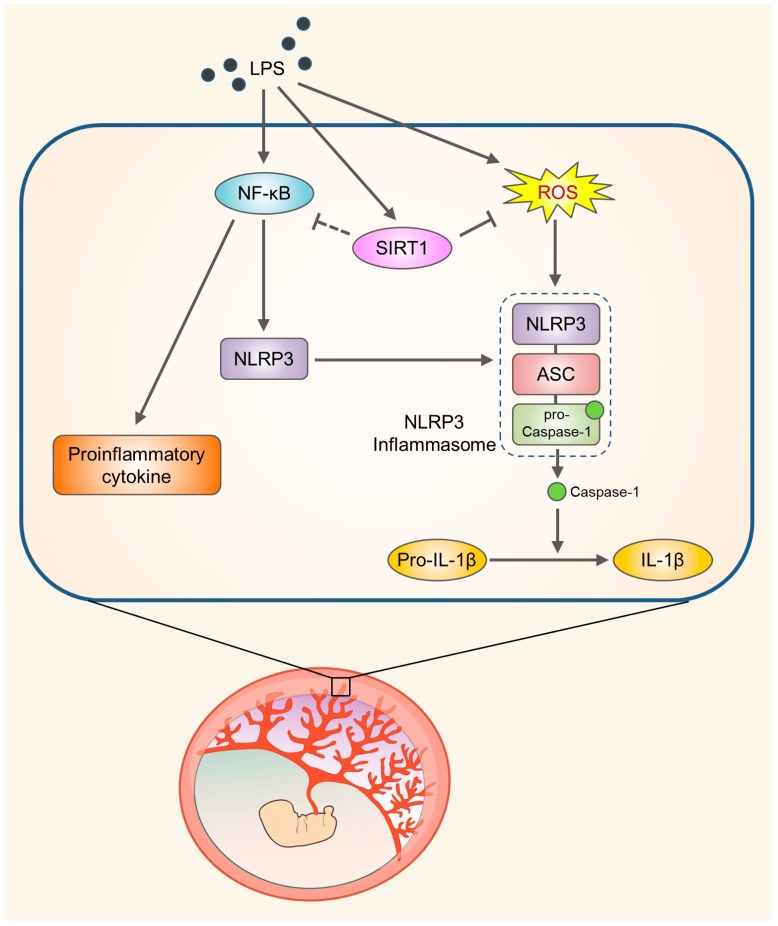
Schematic diagram illustrating the role of SIRT1 in trophoblasts upon exposure to LPS. SIRT1 not only attenuates LPS-induced NLRP3 inflammasome activation but also inhibits inflammatory response and IL-1β secretion by reducing oxidative stress.

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
