# Peer review of "SIRT1 Alleviates LPS-Induced IL-1? Production by Suppressing NLRP3 Inflammasome Activation and ROS Production in Trophoblasts"

_cells, 2020, doi:10.3390/cells9030728_

Round 1

Reviewer 1 Report

Authors of the manuscript have devised and performed in vitro experiments showing SIRT1 is induced in placental tissues of mouse and in the human trophoblasts upon LPS stimulation. They provide experimental evidence that SIRT1 negatively regulates NLRP3 expression and NFkb activation in these cells. The experiments are well-designed, and the data are well presented. However, one major point requires to be addressed with additional data:

The authors claim that SIRT1 inhibits NLRP3 inflammasome activation and to support this claim they show upregulation of NLRP3 mRNA and protein levels and increase in IL-1beta secretion where the SIRT1 is knocked down in the cells. However, NLRP3 inflammasome activation cannot be solely defined by these parameters. Often, especially in murine cells, NLRP3 inflammasome activation requires two signals, PAMPS such as LPS to prime the system (which results in increased transcription of inflammasome components and boosting of NFKb signals) and DAMPS (such as ATP, Alum, …) to induce the activation of inflammasome. This is defined by pro-caspase1 cleavage as well as formation of ASC specks in the cells. So to prove the inflammasome activation in the trophoblasts the authors need to show activated caspase-1 (p10 or p20 subunits by western blot or FLICA assay) or ASC speck formation (by microscopy). Also, the authors cannot claim that the IL-beta measured is certainly mature (bioactive) from. Western blot analysis of IL-1b in the supernatants would help to prove that the released cytokine is the cleaved form (p17). This it remains to be seen if LPS alone is sufficient to activate the NLRP3 inflammasome machinery in this setup or not?

Other points:

Did the authors try to overexpress SIRT1 on the cell lines to see if this would render the cells resistant to NLRP3 inflammasome activation? Going back to the major point raised above on inflammasome activation, did the authors observe decreased expression of other inflammasome components, namely pro-caspase-1 and ASC upon SIRT1 knock down assays? The authors need to provide references (if any) and discuss in the discussion on possible strategies of SIRT1 and also if any SNPs are reported for SIRT1 and its link to inflammatory conditions. In the schematic diagram (Fig. 7) the arrow to show that LPS also induces SIRT-1 expression (via NFKb induction?) is missing.

Reviewer 2 Report

Comments

In this manuscript the Park et al investigated that the role of NLRP3 inflammasome in human trophoblast cells using knockdown of SIRT1 and NLRP3. In the present study, they found that SIRT1 knockdown resulted in hyper-activation of NLRP3 inflammasome and other inflammatory cytokines.

Comments

In the present study, the authors used Sw71 and HTR8 cells and investigated the role of NLRP3 inflammasome in these cells. However, other groups already reported the existence of NLRP3 inflammasome or the mechanisms of NLRP3 inflammasome in these cells (Am J Reprod Immunol.2011 Jun;65(6):542-8. doi: 10.1111/j.1600-0897.2010.00960.x.,  2015;9(5):554-67. doi: 10.3109/17435390.2014.956156., J Reprod Immunol. 2016 Aug;116:104-12. doi: 10.1016/j.jri.2016.06.001. Epub 2016 Jun 6.). The authors have to refer appropriate major literatures. The authors should examine the expression of ASC and caspase-1 because these molecules are essential to activate NLRP3 inflammasome. In addition, Mulla et al. reported that HTR8 cells did not express ASC protein, indicating NLRP3 inflammasome mechanism does not exist in this cell (Am J Reprod Immunol.2011 Jun;65(6):542-8. doi: 10.1111/j.1600-0897.2010.00960.x.). The authors used LPS to stimulate NLRP3 inflammasome, therefore, IL-1b secretion levels were not higher. However, to activate NLRP3 inflammasome, other signals such as ATP, Nigericine, and monosodium urate, are essential. As the present study, LPS actually stimulates NLRP3 and pro-IL1b production, but 2nd signals are needed to stimulate higher levels of IL-1b secretion via NLRP3 inflammasome activation. Therefore, additional experiment are needed.

Reviewer 3 Report

Review Park et al : SIRT1 alleviates LPS-induced IL-1β production by suppressing NLRP3 inflammasome activation and ROS production in trophoblasts

The authors showed a connection between LPS-induced expression o SIRt1 and inflammasome activation both in mouse placena and in human trophoblast cell lines. The methods are adequately described, results are clearly presented.

I have two major concerns with this study:

Firstly, is LPS induction alone sufficient to activate Nlrp3 and increase IL1beta expression? To my knowledge and from what i found in the literature “Activation of NLRP3 inflammasome in macrophages requires two steps: priming and activation.” (Yang et al. Cell Death Dis 2019) 

In this setup, LPS is used solely to indcue inflammasome activation. Please comment.

Related to this issue:

Figure 3B: There is no difference in vehicle treated and LPS induced expression of Nrlp3 mRNA -> shouldnt there be increased expression of Nrlp3 mRNA induced by LPS regardless of SIRT1 knockdown? and then further induction by SIRT1 KD?

Figure 4C: same issue with no or marginal increase in LI-6 and TNFalpha after LPS induction

My second concern is related to the conclusions drawn from Sirt1 knockdown data:

The figure legend for Figure 3 is:  SIRT1 attenuates LPS-induced NLRP3 inflammasome activation in trophoblasts.  -> To draw this conclusion SIRT1 should have been overexpressed and Nrlp3 expression after LPS induction in SIRT1 overexpressing and control cells should have been evaluated. With the present results, the conclusion is: SIRT1 knockdown enhances LPS-induced Nrlp3 mRNA expression

“These data suggest that SIRT1 attenuated the NLRP3 inflammasome activation that was induced by oxidative stress.” Again, this conclusion can not be drawn from the data that are shown. Rather, SIrt1 knockdown increases ROS that is counteracted by NAC. NAC is a control to check whether the effects are ROS-specific, not for SIRT1 action.

There are also some technical issues that I think should be addressed by the authors:

Figure 4B: Please explain/discuss downregulation of p65 phosphorylation at later time points (4 and 6 h)

Figure 6A) ROS is visualized after LPS treatment using CM-H2DCFDA. Please provide a control panel to see that ROS is induced specifically by LPS and visualize cells e.g. by staining of DNA.

Figure 6B) the authors show increased ROS after Sirt1 knockdown. Please also show that ROS production is decreased after SIRt1 overexpression

Round 2

Reviewer 1 Report

The authors have adequately addressed the points I have raised.

It would be more sound if the authors can provide some sort of semi-quantification for confocal images shown in Figure 3F and Figure S4.

Author Response

The authors have adequately addressed the points I have raised.

Response: We thank the reviewer of the positive evaluation of the revised version of our paper. We have further improved the manuscript following the reviewer’s indications, as detailed below.

It would be more sound if the authors can provide some sort of semi-quantification for confocal images shown in Figure 3F and Figure S4.

Response: As suggested by the reviewer, we have quantified endogenous ASC specks and ROS levels and added these data to Figure 3F and Figure S4.

Reviewer 2 Report

The authors have conducted sufficient additional experiments in response to reviewer comments. Numerous experiments have revealed the mechanism by which SIRT1 suppresses the NLRP3 inflammasome.

Minor comment:

1: Please add information about PCR such as accession no. of genes, temperature and time of PCR, and so on.

2: In figure 3 and 6, basal levels of IL-1b secretion was about 10 pg/ml. On the other hand, in figure 5, basal IL-1b secretion was dramatically higher (200 pg/ml) whereas control group. Experiments with the same Sw71 cells, but what is different? Is there any problem as an experimental condition?

Author Response

The authors have conducted sufficient additional experiments in response to reviewer comments. Numerous experiments have revealed the mechanism by which SIRT1 suppresses the NLRP3 inflammasome.

Response: We thank the reviewer of the positive evaluation of the revised version of our paper. We have further improved the manuscript following the reviewer’s indications, as detailed below.

Minor comment:

1: Please add information about PCR such as accession no. of genes, temperature and time of PCR, and so on.

Response: As suggested by the reviewer, we have added extra qPCR information to the Materials and methods sections.

2: In figure 3 and 6, basal levels of IL-1b secretion was about 10 pg/ml. On the other hand, in figure 5, basal IL-1b secretion was dramatically higher (200 pg/ml) whereas control group. Experiments with the same Sw71 cells, but what is different? Is there any problem as an experimental condition?

Response: The reason for this difference is that time of incubation (de la Torre et al., 2009) and the number of the trophoblast cells to collect the conditioned medium was not the same. In Figure 3 and Figure 6, conditioned media were collected from Sw.71 cells 24 h and 34 h, respectively, after LPS or vehicle treatment. On the other hand, in Figure 5B, conditioned media were collected from Sw.71 cells 48 h after LPS or vehicle treatment. In addition, the number of trophoblast cells increased during the double lentiviral infection.

REFERENCE

DE LA TORRE, E., MULLA, M. J., YU, A. G., LEE, S. J., KAVATHAS, P. B. & ABRAHAMS, V. M. 2009. Chlamydia trachomatis infection modulates trophoblast cytokine/chemokine production. J Immunol, 182, 3735-45.

Reviewer 3 Report

The auhors have addressed my concerns with the previous version of the article. 

Author Response

The auhors have addressed my concerns with the previous version of the article.

Response: We thank the reviewer of the positive evaluation of the revised version of our paper.